# Nutritional Education and the State of Oral Health in Adolescents

**DOI:** 10.3390/ijerph19148686

**Published:** 2022-07-17

**Authors:** Małgorzata Jaraszek, Wojciech Hanke, Andrzej Marcinkiewicz

**Affiliations:** Nofer Institute of Occupational Medicine, 91-348 Lodz, Poland; urbaniakmegan726@gmail.com (M.J.); andrzej.marcinkiewicz@imp.lodz.pl (A.M.)

**Keywords:** dental health, diet, lifestyle, dental hygiene, occupational risk factor

## Abstract

**Background.** Health promotion of adolescents is a high priority in Poland. Epidemiological data still indicates the low effectiveness of the health care system in this area. The relationship between the state of oral health and the vocational education of adolescents seems to be justified. The objective is to evaluate changes in the oral health status, eating habits, and the relationship between oral health and lifestyle related to vocational education among students from a secondary technical school of gastronomy and electro-engineering and information technology. **Methods.** The study consisted of two stages (at the beginning and end of education) and two parts: a questionnaire and a dental examination. The study group initially included 55 male students from the secondary technical school of gastronomy and 54 in electro-engineering and information technology school. In the second stage of the study, it was 42 and 51, respectively. **Results.** After three years, in the engineering and information technology group, compared to gastronomy one, a statistically significant increase in caries prevalence and the number of students eating between meals was observed. Conclusions. Professional education on proper nutrition conducted in the gastronomic school could have a decisive impact on the better oral health status of students.

## 1. Introduction

The problem of dental caries (tooth decay) concerns all age groups. In Poland, approximately 80.7% of children aged 12 years, 96% of 18-year-old adolescents, and almost every adult are affected by tooth decay [1]. The highest prevalence of dental caries in Poland among adolescents occurs in the age group of 13–18 years, with a further upward trend in the group ranging between 19 and 34 years [2]. Polish society has been educated for years in maintaining a healthy lifestyle, a balanced diet, and disease prevention. This process takes place during home upbringing, school education, and by means of advertising and educational campaigns carried out in the media [3]. The severity of dental caries depends on diet and lifestyle [4,5]. Despite the advances in knowledge and technology, an effective system of dental caries prevention in different age groups has not yet been established. This justifies the need to constantly search for effective methods of preventing caries development [1,6].

Improper hygienic-, nutritional-, and health-related behaviors of young people may, to a large extent, result from bad home habits and insufficient knowledge related to the level of education at subsequent stages of school education. Most of the students do not treat classes in health promotion seriously. Teachers’ qualifications in oral health promotion are often low, and the theoretical knowledge is too weakly related to its practical application [7]. Failure to combine the theoretical science of healthy eating with its practical application (e.g., as part of practical classes and through meals offered in the school canteen or shop) may result in the failure of students to implement healthy dietary and hygienic habits, which causes the deterioration of the oral health status of those students who do not undergo food and nutrition education training.

The aim of the study was to compare the oral health status of students in the secondary technical school of gastronomy (GA) and the technical school of electro-engineering and information technology (EE–IT) at the beginning of their education and after three years of practical vocational training, and to examine the changes in the eating habits among students of the studied schools, as well as the relationship between the oral health and lifestyle of those continuing their education.

## 2. Materials and Methods

At the first stage of the research, the study group consisted of 109 male adolescents, 55 students from the secondary technical school of gastronomy (GA), and 54 of the secondary technical school of electro-engineering and information technology (EE–IT). The schools were located in the cities of the Łódź province, Poland. All students were 16 years old at the commencement of the study. Ninety-three students (42 from GA and 51 from EE–IT) participated in the second stage of the study after three years of apprenticeship. The reduction in the number of respondents in the second stage resulted from the fact that after the first grade, some students decided to change the professional profile of the school for various reasons.

The criteria for participation in the study were: a resident of the Łódź Voivodeship, aged 15–16, with Polish citizenship, and attending a gastronomic technical school or an electro-mechanical and IT technical school (a secondary school graduate). The criteria of exclusion from participation were: lack of consent of the student or his/her legal guardian to participate in the study, graduation from a school outside the Łódź Voivodeship, or attending a school with a professional profile in the school year preceding participation in the study.

The gastronomic school students were selected for the present study because the specificity of their practical training (cooking, baking and frying, tasting dishes, and preparing confectionery products) favors more frequent food intake and may hinder maintaining proper oral hygiene. The students of the EE–IT school were selected as a group with a completely different specificity of practical apprenticeship, unrelated to food processing.

The examination of each student consisted of two stages and two parts: a questionnaire and a dental examination. The questionnaire concerned individual oral hygiene at home, dental visit frequency, preferred food products, and sources of knowledge about nutrition. A dental examination was performed according to WHO standards, with the use of a mirror and a dental probe, under the light of a dental unit lamp.

The first stage of the study was carried out when students began their education in the first grades. Each respondent was then 16 years old. The second stage of the study consisted of the control examination questionnaire and a dental examination, the same as in the first stage, which was performed after three years of practical vocational training. During the dental examination, the state of oral health, i.e., the number of sound teeth, as well as the “Decayed” (with primary and secondary caries), “Missing,” and “Filled Teeth” (the DMFT Index), and oral hygiene using the Plaque Index (PlI) according to Silness and Loe, were determined.

## 3. Results

Comparing the change in the oral health status after three years of practical vocational training in men from the GA and EE–IT groups, a statistically significant increase in the DMFT index in both groups and an increase in the prevalence of caries (D) in men from the EE–IT group can be seen. However, the opposite tendency is observed in the frequency of fillings (F), which was higher in the GA students in the first and second stages of the study (Table 1).

Table 2 shows the mean values of the bacterial plaque thickness measured by the Plaque Index (PlI) in the GA and EE–IT students in the first and second stages of the study. The students of both the GA and EE–IT groups had a comparable accumulation of plaque on their teeth when they entered the secondary technical school. After three years, a statistically insignificant increase in the Plaque Index value was observed in the students from the EE–IT group and a decrease in the GA group.

At the beginning of the first stage of the study, the students of both groups declared a comparable frequency of tooth brushing. In the second stage of the study, statistically, significantly more students brushed their teeth at least once a day in the EE–IT than in the GA group (Table 3).

Significant differences between students from both schools at the beginning and end of education were revealed by the data on eating habits. In both groups, in the first and second stages of the study, the students consumed a similar number of meals. However, the number of students eating between meals in the second stage of the study was statistically, significantly higher in the EE–IT than in the GA group. Moreover, in comparison to the first stage, there was an increase in the number of students eating between meals in the EE–IT group, whereas a decrease in the GA group was observed.

An unfavorable tendency in the manner of eating meals was also noted in the EE–IT group. In the first stage of the study, meals at fixed times were eaten more often by the students in the EE–IT group. However, after three years, a decrease was recorded in the EE–IT group, while the percentage of students in the GA group eating at fixed times was comparable to that at the beginning of their education.

Students from the gastronomic school more frequently consumed sweets and sweetened hot and cold drinks at the beginning of their apprenticeship. In the second stage, there was a statistically significant increase in the number of students consuming high-carbohydrate products in the EE–IT group, with a simultaneous decrease in the GA students (Table 4).

The school was the main source of knowledge about nutrition among the GA students starting their education (36.4%), followed by the internet (29%) and family (21.8%). Students of the EE–IT school reported their family as the main source of knowledge about nutrition (44.5%). After three years of education in the vocational school, the main sources of nutritional knowledge among the students have changed: it was the school (45.1%) for the GA students and the internet for the EE–IT students (62.5%).

In the first stage of the study, men from the technical gastronomy (GA) and electro-IT schools visited the dentist every 8.2 months on average. In the second stage of the study, the mean time of men from the last visit to the dentist was statistically, insignificantly shortened to 8.1 months in the GA group (*p* = 0.31) and to 7.8 months in the E-I group (*p* = 0.67).

## 4. Discussion

In the first stage of the study, the DMF value in both of the studied groups of students was at a similar level, where GA = 5.29 (minimum = 0, maximum = 12) and EE–IT = 4.59 (where the minimum = 0 and the maximum = 18). These values were also similar to the data concerning the population of 15–16-year-old adolescents in Poland [8]. For comparison, the better situation was diagnosed by Sultan et al. [9]. In a study on the oral health of working Turkish youth, the authors showed that the DMFT values of the students in this group were 2.37 ± 2.45 (minimum = 0, maximum = 13).

After three years of observation, an increase in the DMF value was observed in both groups, which significantly exceeded the data obtained in the general population of 18-year-olds in Poland. The elevated level of the DMF resulted mainly from an increase in the F component (the number of filled teeth) in the GA group and from an increase in the D component (the number of decayed teeth) in the EE–IT group. Comparing the results from the second stage of the study with the data of the 18-year-old adolescents from the Łódź Province, it appears that the surveyed students from the GA school had a comparable number of restored and carious teeth as the 18-year-old adolescents from the Łódź Voivodeship. On the other hand, the EE–IT students had a lower number of restored teeth (by 1.33) and 2.3 more teeth with caries than the 18-year-old adolescents studied by Hilt et al. [10].

In the present study, the examined students from both schools in the first stage of the study were characterized by a comparable level of oral hygiene, PlI = 1.41 (GA) and PlI = 1.28 (EI). Over the three years of observation in the GA group, the quality of oral hygiene improved to a very small, statistically insignificant level, which is comparable to the data in the population of 18-year-olds in Poland [10]. In the EE–IT group, however, the hygiene deteriorated (from PlI = 1.28 to PlI = 1.58; *p* = 0.45), and the results from both stages were higher than the data for the 18-year-old adolescent population in Poland [11].

In the first stage of the study, it was observed that 63.6% of the GA and 63% of the EE–IT students brushed their teeth at least twice a day. About one-fifth of the respondents brushed their teeth once a day, and in each group, nine students declared that they did not brush their teeth at all. These values were comparable with each other but lower than the data presented in the study of 15-year-old adolescents from the Lublin Region, where 70% of the examined adolescents brushed their teeth twice a day and 14% three times a day (after each meal) [12]. After three years, a statistically significant decrease in the frequency of tooth brushing by students from both schools was stated. In the second stage of the study, 30.9% of the GA students (*p* = 0.0014) and only 2% of the EE–IT students (*p* = 0.0000) brushed their teeth at least twice a day. These results are definitely lower than the data in the general population of the country and adolescents from the city of Łódź. The analysis of the literature concerning the frequency of tooth brushing by 16–18-year-old youth from Łódź showed that 89.6% of the surveyed upper-secondary school students performed this activity twice a day and 10.4% once a day [13].

The students from both schools started their education with comparable oral health status and maintained similar oral hygiene care. In the first stage of the study, students from the EE–IT school presented better and healthier eating behaviors, although, in both groups, the students ate their main meals with similar frequency.

In the second stage of the study (after three years), a significant deterioration in eating habits and oral health was observed in the EE–IT group, i.e., in the group where teaching was not related to theoretical knowledge of a healthy diet. In the school of gastronomy, nutrition courses were conducted, contrary to the EE–IT school. It seems that the change in the source of knowledge about nutrition at the beginning and end of their education is the key cause of the differences in the health status and dietary habits between the surveyed groups. Similar trends in changes in eating habits have been described in the literature. Cieślik et al. indicated that students from a gastronomic school derive their knowledge about rational nutrition from school (83.5%), and students of a general upper-secondary school rely on knowledge obtained from the internet (54.5%) [14]. In the literature on the eating habits of adolescents from gastronomic schools, the authors report that despite education in the field of food and nutrition, the respondents declared an irregular consumption of meals and a diet rich in carbohydrates. According to the respondents, habit is the main factor influencing their diet [15]. The way of nutrition of school children is largely connected with the family environment, which, according to some authors, may result from different value systems, lifestyles, and the method of supplying food. Apart from the family, school is the second educational environment in which it is possible to affect the nutritional behavior of children and students during classroom activities and through meals offered in the school canteen, shop, or boarding house. The author reported a better diet for female students from gastronomic schools who, during the school year, lived in a boarding house than those living at home [16].

The literature reports that the content and form of classes, as well as the food offered at school, can affect the nutritional behavior of students. It is likely that these factors may also influence and develop other elements of a healthy lifestyle, e.g., eating style and proper oral hygiene [16].

However, in the students’ opinion, the forms of educational activities are unattractive and provide little new information. They point out that teachers who have taken only courses preparing them to conduct classes in health promotion are often not credible and competent enough to teach students properly [7]. Taking into account all these problems and the analysis of the results obtained in this study, it might be right to employ professional health educators, e.g., dental hygienists and nutritionists, to work in upper-secondary schools. Classes conducted in an attractive form of workshops or training would allow students to develop effective mechanisms of healthy habits [7].

About 70% of environmental conditions, as well as hygienic and nutritional habits, determine the intensity of caries. In highly developed countries, an increase in the number of people free from caries has been observed for about 40 years. However, in Poland, there is a delay compared to Western European countries. Caries is still a serious dental problem [17]. Within the so-called intentional education, pupils’ eating habits can be changed. Therefore, a lack of relationship between the theoretical knowledge of healthy eating and its practical application may result in deterioration in the oral health of students whose vocational education is not related to food and nutrition. Students from both schools (GA and EE–IT) entered upper secondary school with a comparable state of oral health, better than the population of 15–16-year-olds in Poland. They maintained a similar level of oral hygiene. In both schools, students ate a similar number of main meals and ate between meals with similar frequency. However, after three years, a significant deterioration in their eating habits and oral health was observed in the EE–IT group, i.e., in the group where training was not associated with theoretical knowledge on healthy eating. Healthcare professionals play an important role in promoting the healthy lifestyle of children and adolescents [7,18]. Perhaps it would be reasonable to employ professional health educators, such as dental hygienists and nutritionists, to work in schools. Classes performed in an attractive form would allow students to develop effective mechanisms of healthy habits [7,18].

## 5. Conclusions

Professional education concerning proper nutrition in the school of gastronomy has a decisive impact on a better state of oral health.Lack of professional education on nutrition among adolescents in which the internet is the main source of knowledge in this field may significantly contribute to the deterioration of oral health.To improve oral health among adolescents and young adults, the following issues should be considered: the improvement of the quality and availability of education on oral hygiene, rational nutrition, and a healthy lifestyle at school and the dental office.Employment of professional educators, such as dental hygienists and nutritionists, in upper-secondary schools to improve the effectiveness and attractiveness of health- promoting classes is recommended.The study requires further observation of the larger population of the country and in other fields of vocational education.

## Figures and Tables

**Table 1 ijerph-19-08686-t001:** Mean values of the DMF index and its components among the students of the gastronomic group (GA group) and electro-engineering and information technology school group (EE–IT group) in the first and second stages of the study.

	Stage I			Stage II			*p* Value (Difference between Stage I and II)	
	GA Group *n* = 55	EE–IT Group *n* = 54	*p* Value	GA Group *n* = 42	EE–IT Group *n* = 51	*p* Value	GA Group	EE–IT Group
DMF index	5.29	4.59	0.098	7.35	7.17	0.45	0.012	0.0075
D (decay)	1.76	2.11	0.49	2.57	3.4	0.16	0.027	0.011
F (filled)	3.23	2.23	0.0029	4.55	3.7	0.054	0.03	0.026
M (missing)	0.3	0.02	0.13	0.28	0.07	0.27	0.5	0.3

**Table 2 ijerph-19-08686-t002:** Mean values of the Plaque Index (PlI) in the GA and EE–IT students in the first and second stages of the study.

	Stage I			Stage II			*p* Value (Difference between Stage I and II)	
	GA Group *n* = 55	EE–IT Group *n* = 54	*p* Value	GA Group *n* = 42	EE–IT Group *n* = 51	*p* Value	GA Group	EE–IT Group
Plaque Index (PlI)	1.41	1.28	0.18	1.34	1.58	0.34	0.32	0.45

**Table 3 ijerph-19-08686-t003:** The frequency of tooth brushing declared by the GA and EE–IT students in the first and second stage of the study.

	Stage I			Stage II			*p* Value (Difference between Stage I and II)	
The Frequency of Tooth Brushing	GA Group *n* = 55	EE–IT Group *n* = 54	*p* Value	GA Group *n* = 42	EE–IT Group *n* = 51	*p* Value	GA Group	EE–IT Group
Twice a day or more	35 (63.6%)	34 (63%)	0.94	13 (30.9%)	1 (2%)	0.0001	0.0014	<0.00001
Once a day or less often	11 (20%)	11 (20.4%)	0.96	27 (64.3%)	49 (96%)	0.000079	0.00001	0.00001
No brushing	9 (16.4%)	9 (16.6%)	0.96	2 (4.8%)	1 (2%)	0.45	0.07	0.01

**Table 4 ijerph-19-08686-t004:** Nutritional behavior of the GA and EE–IT students in the first and second stages of the study.

	Stage I			Stage II			*p* Value (Difference between Stage I and II)	
Nutritional Behavior	GA Group *n* = 55	EE–IT Group *n* = 54	*p* Value	GA Group *n* = 42	EE–IT Group *n* = 51	*p* Value	GA Group	EE–IT Group
Mean number of meals eaten during the day	4	4.3	0.33	3.9	4.2	0.51	0.39	0.89
Number of students eating between meals	24 (43.6%)	16 (29.6%)	0.13	10 (23.8%)	38 (74.5%)	<0.00001	0.043	<0.00001
Number of students at fixed times	8 (14.5%)	16 (29.6%)	0.057	10 (23.8%)	3 (5.9%)	0.013	0.24	0.0016
Number of students eating sweets and drinking hot and cold sweet drinks	44 (80%)	34 (63%)	0.048	30 (71.4%)	46 (90.2%)	0.02	0.32	0.0011

## Data Availability

Some or all data and models that support the findings of this study are available from the corresponding author upon reasonable request.

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
