# Peer review of "Nutritional Education and the State of Oral Health in Adolescents"

_ijerph, 2022, doi:10.3390/ijerph19148686_

Round 1

Reviewer 1 Report

Thanks for this paper.

I would recommend to improve the conclusion section. Here the author discuss the GA grupo but not the other one. I believe more could be added here taking advantage of the resutls.

Also, I would like to see a future reaserch secction and/or a reseach implications for wider population. 

With these minor changes I would recomend for publication. 

Thanks.

Author Response

Thank you very much for the review and comments, thanks to which the paper will be able to become more scientifically valuable. The conclusions section has been enriched with a discussion of the results concerning the E-I group. A note on the implications of the research for the wider population was also added.

Reviewer 2 Report

Congratulation for the wonderful paper. Thank you for this helpful contribution. I applaud the effort of promoting studies for the investigation of nutritional education. I appreciate their methods including study design and data analysis. I write some comments below that could benefit the article.

Results. Tables. Information is shown in duplicate. If it appears in the table, delete the information from the text.

Conclusion. Conclusion must respond to the main aim of the study. It is advisable that conclusion be clear and concise. Therefore, it is not recommended that its length be so long.

References. Books are not science. It is advisable to reference articles published in scientific journals.

Congratulation again!!!

Author Response

Thank you very much for the review and comments, thanks to which the paper will be able to become more scientifically valuable. Changes in the text regarding tables, results and conclusions have been introduced in the text. I agree that it is advisable to reference articles published in scientific journals. As you suggested, the number of references-books was reduced to one. The fragment of the book used in the publication significantly describes the phenomenon discussed in the text.

Reviewer 3 Report

The present study set out to compare the oral health status of students in the secondary technical school of gastronomy and the school of electric engineering and information technology, and its relation to the lifestyle of those continuing their education.

The topic of study is interesting from a clinical standpoint.

There are some comments below:

1.     You have to explain clear the inclusion and exclusion criteria.

2.     In my viewpoint, the second stage of the study should be included a dental behaviour questioner to investigate the tooth brushing frequency, dental visit frequency during these three years. These should be explained in materials and methods part.

3.     How performed the oral examination? Please, explain.

4.     Do you have ethical approval?

Author Response

Thank you very much for the review and comments, thanks to which the paper will be able to become more scientifically valuable.

  1. The criteria for participation in the study: a resident of the Lodzkie Voivodship, aged 15-16, with Polish citizenship, signed up in a gastronomic technical school or an electro-mechanical and IT technical school (immediately after graduating from gymnasium).
    The criteria of exclusion from participation: lack of consent of the student or legal guardian to participate in the study, graduation from gymnasium outside the Lodzkie Voivodeship, attending a school with a professional profile in the school year preceding participation in the study.
  1. The material and methods were supplemented with information on the course of the questionnaire survey in the second stage.
  2. An examination of the oral cavity was carried out in the school dentist's office. The teeth were viewed in the illumination of the dental unit lamp. Using a disposable dental mirror and probe. A carious lesion was classified as lesions within the enamel and dentin, where the tip of a dental probe can go inside.
  3. Yes. The study received approval from the bioethics committee on September 22, 2015.

We have included all the changes in the article.